# Bidirectional regulation of the brain-gut axis in *Macaca mulatta*: implications for wildlife conservation and experimentation

Zewen Sun,[1,2,3] Jun Wang,[1,3] Ruiping Sun,[1] Baozhen Liu,[4] Keqi Cai,[5] Xinyuan Zhao,[4] Yanfang Wang,[3,6] Jianguo Zhao,[2,3] Jingli Yuan[1,3]

**ABSTRACT** Although existing research has uncovered the association between psychological stress and gut microbiota dysbiosis, the causal relationship remains unclear. The direct impact of psychological stress on gut microbiota and the potential bidirectional mechanisms remain unclear, including the specific molecular pathways involved. This study investigates the impact of psychological stress on the gut microbiota and associated metabolites in wild Hainan macaques, revealing that stress significantly alters microbial composition and function. Specifically, stress-induced changes in the gut microbiota are linked to shifts in key metabolites, particularly coumarins, which are known to interact with the nervous system. This suggests a feedback loop where stress modulates neurological function via microbiota-derived metabolites. We identified several differential metabolites, including specific C10757, that can serve as biomarkers for detecting stress-induced health risks. These findings highlight the potential of microbiota-based interventions to mitigate stress-related health issues and provide essential data for wildlife health monitoring. The study highlights the role of gut microbiota as a stress biomarker, underscoring the importance of psychological well-being in wildlife conservation and research to guide ethical animal management.

**IMPORTANCE** This study uncovers how psychological stress alters gut microbiota in wild animals, enhancing understanding of the gut-brain axis in natural ecosystems. Crucially, it identifies microbial profiles as non-invasive stress biomarkers, enabling early detection of environmental threats in conservation. The findings emphasize the ethical need to incorporate stress assessments (e.g., microbiota and cortisol analyses) into wildlife research to ensure welfare and data validity. Mechanistic parallels with captive species suggest evolutionary conservation of gut-brain pathways, opening avenues for cross-species therapies. By bridging psychobiology and conservation, this work establishes a framework for stress resilience interventions and ethical wildlife management, advancing both ecological health and humane science. Future research should explore microbiota-targeted strategies and cross-species applicability to optimize conservation outcomes.

**KEYWORDS** gut microbiota, brain-gut axis, *Macaca mulatta*, wildlife conservation, experimentalization

In recent years, an increasing number of studies have unveiled the complex interplay between mental stress and the gut microbiota (1–3). Both in human and animal models, short-term or chronic mental stress has been shown to significantly alter the structure of gut microbial communities, thereby affecting the host's physiological and psychological health (4). For instance, Chang et al. demonstrated that psychological stress, by impacting the function of Brunner's glands in the gut, leads to microbial imbalance, particularly a reduction in *Lactobacillus* spp., ultimately triggering intestinal inflammation and a decline in immune function (5).

Address correspondence to Jingli Yuan, 13250732023@163.com, or Jianguo Zhao, zhaojg@ioz.ac.cn.

Zewen Sun and Jun Wang contributed equally to this article. Author order was determined by drawing straws.

The authors declare no conflict of interest.

See the funding table on p. 18.

The gut-brain axis is a complex bidirectional communication system between the gut and the brain, involving multiple pathways including neural, endocrine, immune, and metabolic mechanisms (6, 7). Gut microbes influence the central nervous system via the vagus nerve and the enteric nervous system, regulating emotions and behavior (8, 9). They also affect the hypothalamic-pituitary-adrenal axis by modulating the release of hormones from enteroendocrine cells, thereby regulating stress responses (10). Moreover, gut microbes impact systemic inflammation and brain function by modulating the immune system and producing metabolites such as short-chain fatty acids and tryptophan metabolites (11, 12).

Most current studies rely on rodent models, which exhibit significant physiological and behavioral differences from humans, limiting the direct extrapolation of their findings (13, 14). In contrast, non-human primates, being close relatives of humans, share more similar physiological, immunological, and behavioral characteristics, making them ideal models for studying human-related diseases and physiological processes (15–17). Moreover, wild non-human primates develop unique gut microbiota structures in their natural habitats, and their lack of domestication allows for a better simulation of the human physiological state in natural environments (18, 19). The Hainan subspecies of *Macaca mulatta* represents a critically important wild primate model due to its purely undomesticated status and ecological isolation in tropical forests (20). Unlike captive-bred conspecifics, this subspecies exhibits natural behavioral repertoires and undisturbed gut microbiome profiles (21), providing a unique opportunity to study stress-microbiome interactions in an evolutionary relevant context. Recent metagenomic studies have documented distinct microbial signatures in wild Hainan macaques compared to captive populations (22), though the functional implications of these differences remain unexplored, a gap this study directly addresses. When these wild monkeys are introduced to captive environments, the changes in their living conditions can simulate the psychological stress scenarios faced by humans in modern society, providing a more realistic and reliable model for studying the impact of psychological stress on the gut microbiota (23, 24).

This study employs a large cohort of wild, untrained Hainan subspecies of rhesus macaques as subjects. By subjecting wild macaques to captivity-induced psychological stress through restricted movement, we aim to investigate the effects of psychological stress on their gut microbiota and the regulatory role of the microbiota and its metabolites on brain function. By comparing the gut microbiota structure of captive and wild macaques, we seek to elucidate how psychological stress alters microbial communities and further explore the potential physiological and behavioral consequences. This research not only deepens our understanding of the interplay between psychological stress and the gut microbiota but also provides a new perspective for developing gut microbiota-based intervention strategies to address the increasingly prevalent stress-related diseases in modern society.

## RESULTS

### Identification of the gut microbiota under different psychological stress levels

To establish an animal model of psychological stress, we selected wild-born, unhabituated rhesus monkeys and provided them with a captive breeding environment in collaboration with a wildlife conservation organization. Following captivity, these monkeys exhibited significant depression or emotional instability (Fig. 1A). We assembled the quality-controlled data separately for each sample (Table S1; Fig. S1A). There were no significant differences between the wild and captive groups in terms of Chao1, Shannon, and Simpson indices, indicating that both groups had high gene richness, evenness, and gene diversity (Fig. S1B). Additionally, there were no significant differences between the samples of the wild and captive groups, and the dispersion of the samples was within an acceptable range (Fig. S1C). Further analysis revealed that the species accumulation curves of both the wild and captive groups showed an upward trend at the

end and tended to flatten, indicating that the sampling volume was sufficient to reveal the species diversity of the community (Fig. S1D). The Chao1 index of the wild group was significantly higher than that of the captive group ($P = 0.041$), indicating that wild macaques had more microbial species in their feces compared to captive macaques (Fig. S2A). However, there were no significant differences across groups in the Shannon and Simpson indices, indicating that the differences in species evenness between wild and captive macaques were not significant (Fig. S2B). Moreover, it was found that there were significant differences in the microbial community composition between the wild and captive groups (Fig. S2C and D), with extremely significant differences between the two groups ($R^2 = 0.1752$, $P = 0.008$, Fig. S2E). Taken together, captive-induced psychological stress led to behavioral disturbances and significant gut microbiota alterations, including reduced microbial richness and distinct community composition.

## The alterations in the gut microbiota caused by different psychological stress levels

To clarify stress-induced gut microbiota changes, we characterized intergroup microbial composition differences. After species annotation, we obtained 8,702 species annotations, including 7,653 bacterial species, 409 archaeal species, 142 fungal species, and 377 viral species. We analyzed the community structure of the fecal microbiota of macaques in the wild and captive groups at the phylum, genus, and species levels. At the phylum level, the fecal microbiota of macaques in both the wild and captive groups was mainly dominated by Firmicutes, Bacteroidetes, and Proteobacteria (Fig. 1B). At the genus level, the fecal microbiota of macaques in both groups was primarily composed of *Faecalibacterium*, *Prevotella*, and *Treponema* (Fig. 1C). At the species level, the fecal microbiota of macaques in both groups was mainly dominated by *Faecalibacterium_prausnitzii*, *Treponema_succinifaciens*, and *Dysosmobacter_welbionis* (Fig. 1D). Analysis of the differences in fecal microbial species of the groups revealed that there were 4,293 common microbial species, with 491 unique species in the wild group and 250 unique species in the captive group (Fig. 1E). Using the linear discriminant analysis (LDA) effect size (LEfSe) test ($P < 0.01$, LDA > 2), we analyzed the differences in abundance between the wild and captive groups at the genus and species levels of the gut microbiota. The results showed that there were 100 genera and 160 microbial species with significant differences between the two groups. The dominant genera in the captive group were *Odoribacter*, *Ornithobacterium*, and *Enterobacter*, while the dominant genera in the wild group were *Pasteurella* and *Succinivibrio* (Fig. 1F). At the species level, the wild group exhibited significantly higher abundance of bacteria involved in host dietary digestion and energy metabolism, particularly cellulose-degrading and short-chain fatty acid-producing taxa (e.g., Faecalibacterium prausnitzii and Succinivibrionaceae Aeromonadales). In the captive group, the bacteria with significantly higher abundance, which were related to the host's dietary digestion and energy metabolism, capable of fermenting various organic substances and producing short-chain fatty acids, included *Lactobacillus animalis* and *Intestinimonas_butyriciproducens* (Fig. 1G; Fig. S2F). Collectively, our findings indicate distinct gut microbiota profiles between wild and captive macaques, with captive specimens showing reduced microbial diversity, while wild populations maintained higher abundances of fiber-degrading species linked to plant-rich diets.

## Gut microbial functional abnormalities caused by psychological stress

To investigate the impact of different psychological stress levels on the functional distribution of gut microbiota in macaques, we performed functional annotation on the non-redundant genes predicted from the fecal microbiota of captive and wild macaques. Functional enrichment analysis revealed that the metabolic pathways in both groups were primarily enriched in global and overview maps, carbohydrate metabolism, amino acid metabolism, membrane transport, and metabolism of cofactors and vitamins (Fig. 2A). Based on Bray-Curtis distance, the genes in the captive and wild groups formed distinct clusters (Adonis test, $R^2 = 0.08$, $P = 0.04$), indicating significant differences

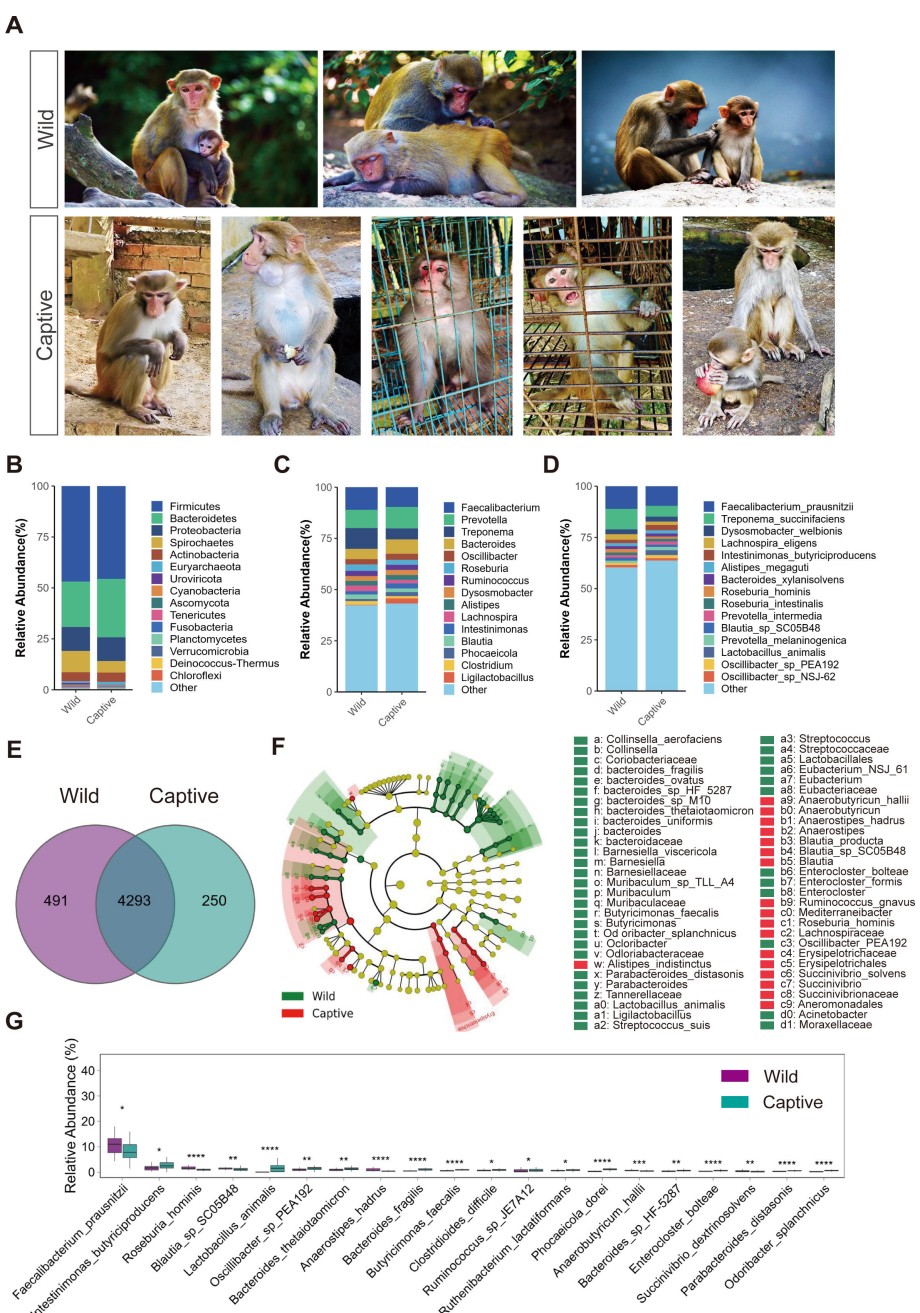

**FIG 1** Differential analysis of fecal microbiota in macaques. (A) Representative images of macaques in the wild and captive groups. (B–D) Stacked bar charts of species abundance at the phylum (B), genus (C), and species (D) levels. The *x*-axis represents the groups, and the *y*-axis represents the relative abundance of species. The colors of the bars indicate the taxonomic classification of the species. (E) Venn diagram at the species level. The numbers in the overlapping regions represent the number of species shared among the groups, while the numbers in the non-overlapping regions indicate the number of species unique to each group. (F) Linear discriminant analysis effect size cladogram at the genus level. Each dot represents a specific taxonomic classification, with the size of the dot indicating the relative abundance. Significant differences are colored by group, and the colors of the pie charts represent higher-level taxonomic classifications. (G) Box plots of differential species abundance at the species level. The *x*-axis represents the species with statistically significant differences, and the *y*-axis represents the relative abundance of the species. *$P<0.05$; **$P<0.01$; ***$P<0.001$.

between the two groups (Fig. 2B). The abundance of metabolic pathways based on functional annotation changed, with particularly noticeable differences in the pathways with the highest relative abundance. High-abundance pathways such as K07133 and

K21572 exhibited higher abundance in the captive group (Fig. 2C). To further explore the specific functional differences between the two groups, we conducted enrichment analysis using the module database for the captive and wild groups. The results showed that the gut microbiota of the wild group had higher gene enrichment in 11 pathways, including sphingolipid biosynthesis; and globin and globin-like proteins; methane metabolism; O-antigen nucleotide sugar biosynthesis; alanine, aspartate, and glutamate metabolism; cysteine and methionine metabolism; and amino sugar and nucleotide sugar metabolism, compared to the captive group. In contrast, the gut microbiota of the captive group had higher gene enrichment in five pathways, including photosynthesis, phosphate and phosphonate metabolism, butanoate metabolism, atrazine degradation, and secondary bile acid biosynthesis, compared to the wild group (Fig. 2D). Therefore, captivity significantly affects the metabolic functions of the gut microbial community in macaques, with pronounced differences in abundance observed in several key metabolic pathways. Together, captivity induces significant metabolic reprogramming in macaque gut microbiota, with wild populations showing enrichment in sphingolipid and amino acid metabolism pathways, while captive specimens exhibit upregulation of butanoate metabolism and secondary bile acid biosynthesis.

## Metabolic and antibiotic resistance alterations caused by psychological stress

To investigate how psychological stress influences host metabolism and biological functions through gut microbiota, we conducted further research. Carbohydrate metabolism, a vital pathway for host energy metabolism, exhibits high gene enrichment in related metabolic pathways. Principal coordinate analysis (PCoA) based on Bray-Curtis distance revealed that the composition of carbohydrate-active enzymes (CAZymes) in captive and wild macaques formed distinct clusters, indicating differences across groups (Fig. 3A). Using the Carbohydrate-Active Enzymes (CAZy) database, we predicted the composition of CAZymes in the gut microbiota of macaques. After classification and annotation, the major CAZymes were identified as glycoside hydrolases (GHs), glycosyl-transferases (GTs), carbohydrate-binding modules (CBMs), carbohydrate esterases (CEs), polysaccharide lyases, and auxiliary activities (Fig. 4B). Statistical Analysis of Metage-nomic Profiles (STAMP) analysis was employed to screen for the top 30 most abundant CAZymes with significant differences between the captive and wild groups. In the captive group, the abundance of genes encoding 21 glycoside hydrolases (e.g., GH13, GH3, GH97), 2 carbohydrate-binding modules (CBM20, CBM32), 2 carbohydrate esterases (CE0, CE11), and 3 glycosyltransferases (GT4, GT30, GT3) was significantly higher than in the wild group ($P < 0.05$; Fig. 3B). Conversely, the abundance of genes encoding two glycoside hydrolases (GH1, GH43_24) and one carbohydrate-binding module (CBM13) was significantly higher in the wild group than in the captive group ($P < 0.05$, Fig. 3C). This suggests that captive macaques possess a richer gene repertoire for carbohydrate-active enzymes compared to wild macaques.

PCoA based on Bray-Curtis distance also showed that the composition of antibiotic resistance genes in captive and wild macaques formed distinct clusters, indicating differences between the two groups (Fig. 3D). Comparison with the Comprehensive Antibiotic Resistance Database (CARD) revealed that the most abundant functional genes annotated were related to tetracycline, macrolide, aminoglycoside, cephalosporin, and fluoroquinolone antibiotics (Fig. 3E). Statistical analysis of the relative abundance (transcripts per million [TPM] values) of these genes showed that the abundance of tetracycline and macrolide resistance genes was significantly higher in the captive group than in the wild group ($P < 0.05$). In contrast, the abundance of aminoglycoside, penicil-lin, and fluoroquinolone resistance genes was significantly higher in the wild groups than in the captive groups ($P < 0.05$, Fig. 3F). In summary, captive macaques exhibit distinct gut microbial carbohydrate metabolism profiles with significantly enriched glycoside hydrolase genes and elevated tetracycline/macrolide resistance genes compared to wild counterparts, reflecting diet-driven microbial adaptation.

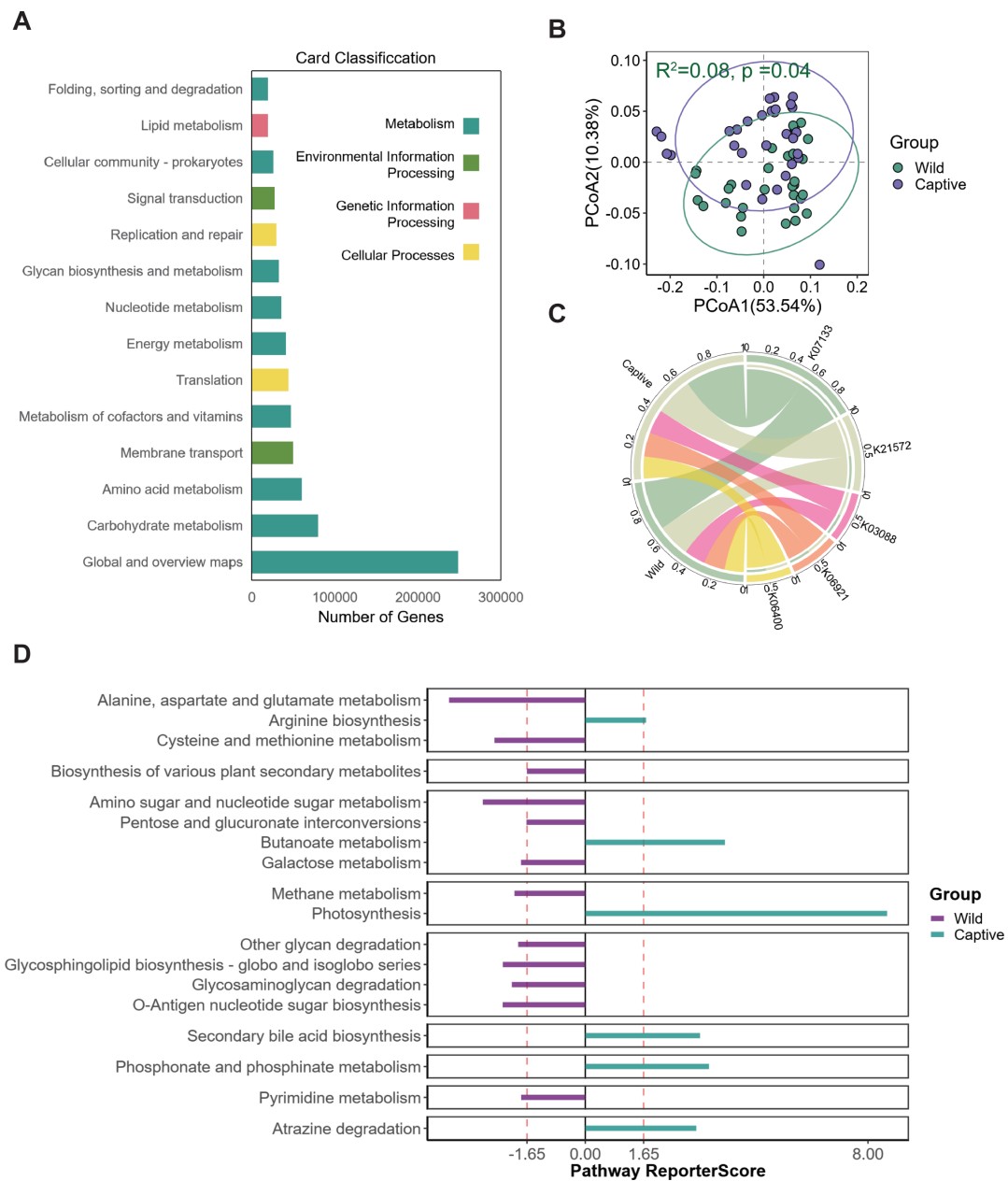

**FIG 2** Differential analysis of fecal microbiota function in macaques. (A) Bar chart of functional gene counts. The *x*-axis represents the number of genes, and the *y*-axis represents functional categories. The color of the bars indicates the group. (B) Principal coordinate analysis (PCoA) plot of gut microbiota functional genes. (C) Circos plot of differential functions. The left side of the circle indicates group information, and the right side indicates functional classification. The outermost scale represents the proportion of functions in the samples or the proportion of different functions within a sample. The color of the inner segments indicates the group, and the presence of a connection between segments indicates the presence of that function in the sample. The width of the arc represents the proportion. (D) Kyoto Encyclopedia of Genes and Genomes (KEGG) pathway enrichment difference plot. The *x*-axis represents the reporter score value, and the *y*-axis represents the pathway. The dashed line indicates the significance level of the reporter score.

## Metabolite composition differences under different stress environments

We further conducted untargeted metabolomics analysis on fecal samples from captive and wild macaques to investigate the differences in metabolite composition under different stress environments. Qualified samples were subjected to metabolite profiling (Fig. 4A). All detected metabolites were enumerated(Fig. 4B) and categorized by their primary metabolic pathways (superpathway). Lipid metabolites were further classified by

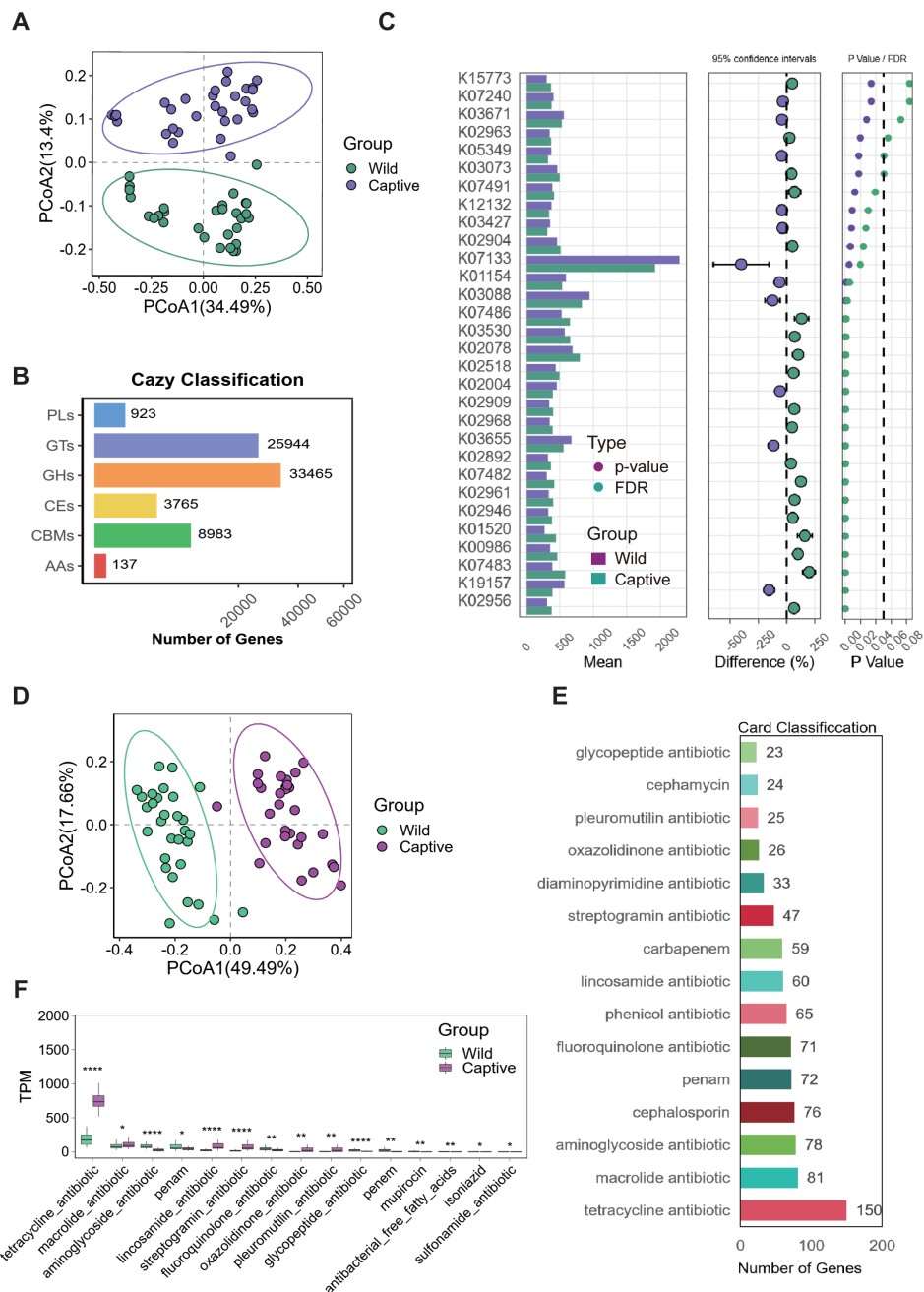

FIG 3 Differential analysis of CAZymes and antibiotic resistance. (A) PCoA plot of gut microbiota CAZymes. (B) Bar chart of functional gene counts of CAZymes in macaque gut microbiota. (C) Differences in CAZymes between the wild and captive groups (STAMP analysis, $P < 0.05$). (D) PCoA plot of antibiotic resistance genes in gut microbiota. (E) Bar chart of functional gene counts of antibiotic resistance genes in macaque gut microbiota. (F) Differences in antibiotic resistance genes between the wild and captive groups ($P < 0.05$). *P<0.05; **P<0.01; ***P<0.001.

sub-pathway (Fig. 4C). The results showed that among all fecal metabolites in macaques, the highest number of metabolites belonged to amino acid metabolism, with a total of 121 species, accounting for 22.92% of the total metabolites. Biosynthesis of other secondary metabolites had 98 species (18.56%), and lipid metabolism had 74 species (14.02%). Statistical analysis of the relative abundance of metabolites in the captive and wild groups revealed that the most abundant metabolites were amino acids, peptides and their analogs, and fatty acyls (Fig. 4D). The findings indicate that amino acid metabolism and biosynthesis of other secondary metabolites are the most dominant

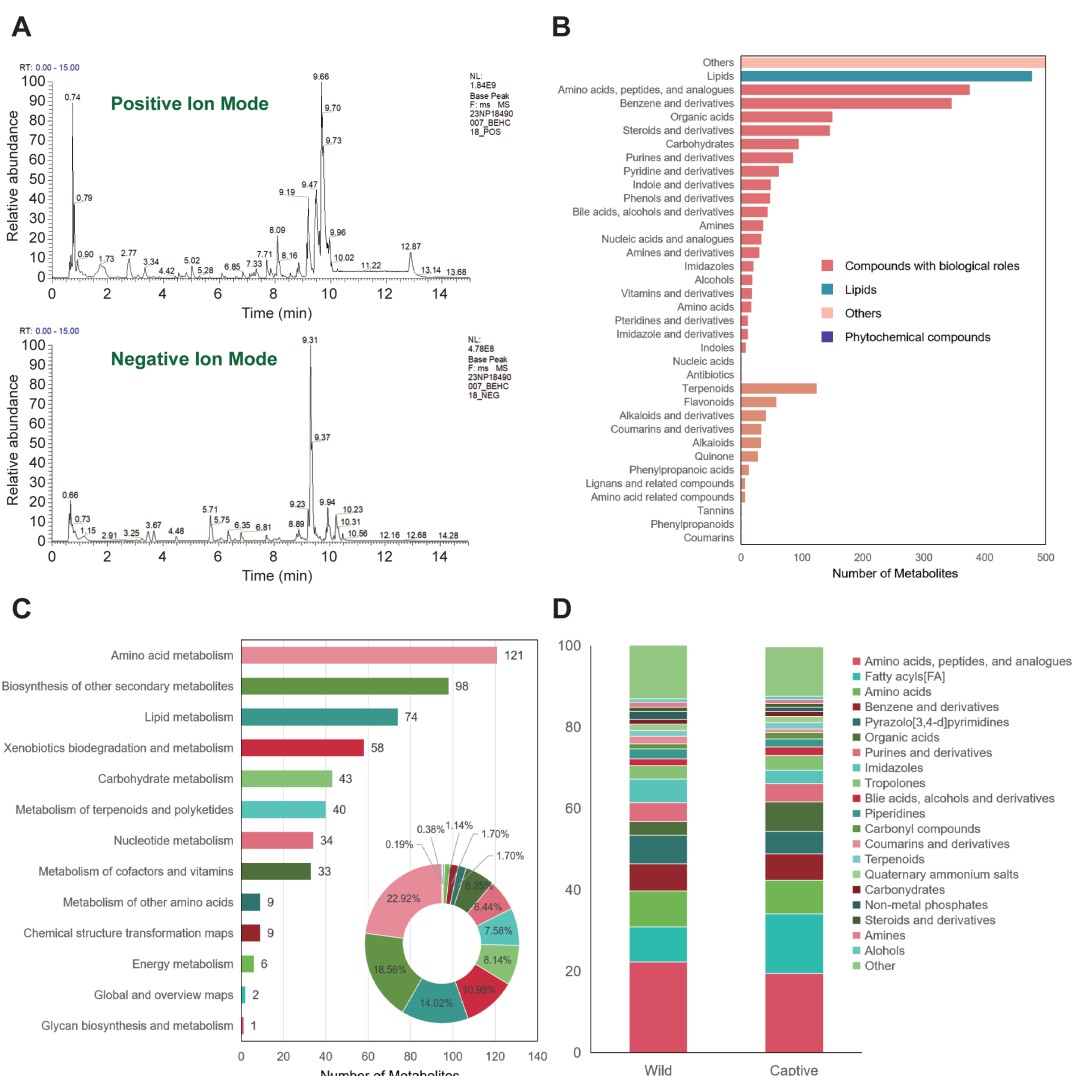

**FIG 4** Classification of fecal microbial metabolites in macaques. (A) Base peak chromatograms. The left panel shows the positive ion mode, and the right panel shows the negative ion mode. The *x*-axis represents retention time, and the *y*-axis represents the response intensity of the most intense ion detected at each time point. (B) Bar chart of metabolite final class classification. The *y*-axis indicates the class to which the metabolites belong, and the *x*-axis indicates the number of metabolites. (C) Kyoto Encyclopedia of Genes and Genomes (KEGG) classification of metabolic pathways. The *y*-axis indicates the class to which the metabolic pathways belong, and the *x*-axis indicates the number of metabolites. Different colors represent different metabolic pathway classification entries, and the percentages indicate the proportion of metabolites in each class relative to the total number of metabolites with pathway information. (D) Stacked bar chart of common metabolite classification composition (top 20 most abundant metabolites). The *y*-axis indicates the proportion of metabolites.

metabolic pathway categories in macaque fecal metabolites. This likely reflects the characteristics of gut microbial metabolism and host-microbiota interactions, providing an important basis for further research into their physiological functions and metabolic features.

## Psychological stress leads to altered gut metabolite composition

To further clarify whether psychological stress alters gut metabolites, we analyzed the fecal metabolic composition using metabolomics. Firstly, we established principal component analysis (PCA) models for the wild and captive groups and validated them using permutation tests, finding that the samples in both groups clustered tightly and had high correlation, indicating minimal systematic error and good reproducibility of the experiments (Fig. 5A and B). Subsequently, we conducted univariate analysis to compare differential metabolites in the feces of the wild and captive groups. The results

showed that, compared with the wild group, the captive group had 999 metabolites with significantly increased levels and 959 metabolites with significantly decreased levels (Fig. 5C and D). To further analyze the expression patterns of these differential metabolites, we performed clustering analysis on their expression levels. The results revealed that a total of 11,958 differential metabolites in the fecal samples of the two groups exhibited significant intergroup differences in expression levels, and these differential metabolites showed consistent expression patterns within the samples of the wild and captive groups, respectively (Fig. 5E). To more intuitively reveal the synergistic or antagonistic relationships between different types of differential metabolites, we constructed a chord diagram of differential metabolite correlations based on the Spearman correlation coefficients between metabolites ($|r| > 0.8$ and $P < 0.05$). The results showed that the expression levels of coumarins and derivatives and 1,4-dioxanes were negatively correlated, while the remaining groups generally exhibited positive correlations (Fig. 5F). To further explore the differences in high-abundance metabolites across groups, we analyzed the 20 metabolites with the largest differences between the groups. The results showed that the metabolites with high expression in the wild group included the human metabolome database (HMDB) 0029218 (urolithin C), 9.062_417.29982, and 9.028_315.19611 (15-deoxy-12,14-prostaglandin A2), while the metabolites with higher expression levels in the captive group included 3.512_206.11780 (2-piperidinobenzoic acid), PB00581 (angioletin), and HMDB0002085 (syringic acid) (Fig. 5G and H). These results demonstrate profound gut metabolite alterations, including stress-associated compounds like syringic acid (captive-enriched) and anti-inflammatory urolithin C (wild-enriched).

## Metabolite functional alterations caused by different stress environments

We performed metabolic pathway enrichment analysis on the differential metabolites, and the results showed that many differential metabolites were enriched in pathways such as dopaminergic synapse, retrograde endocannabinoid signaling, bile secretion, and steroid hormone biosynthesis ($P < 0.01$, Fig. 6A). To further analyze the overall changes of these differential metabolites in the pathways, we analyzed the top 10 significantly enriched metabolic pathways in the two groups. The results showed that all metabolites in the alanine, aspartate, and glutamate metabolism, alcoholism, amphetamine addiction, and cocaine addiction pathways were significantly upregulated in the captive group. In contrast, all metabolites in the linoleic acid metabolism and retrograde endocannabinoid signaling pathways were significantly downregulated in the captive group. In addition, the overall expression levels of metabolites in the dopaminergic synapse pathway were upregulated, while those in the steroid hormone biosynthesis, tyrosine metabolism, and bile secretion pathways were downregulated (Fig. 6B; Table S2). Finally, we conducted receiver operating characteristic (ROC) analysis on the differential metabolites, and the results showed that the area under the curve (AUC) value of metabolite C10757 was 0.999, indicating that this metabolite has high discriminability and is suitable as a biomarker to distinguish between wild and captive macaques (Fig. 6C). To reveal the sources of differential metabolites and the metabolic pathways through which microbes produce the relevant metabolites, we used the MetOrigin platform to trace the sources of differential metabolites. The results showed that 126 metabolites were annotated to the KEGG and HMDB databases: four were from the macaque host, 25 were derived from gut microbes, 33 were co-metabolites of microbes and host, and the remaining 207 metabolites were from other sources (Fig. 6D). Based on metabolic source pathway enrichment, there were 5, 24, and 50 enriched pathways for host, microbial, and microbial-host co-metabolism, respectively. Among them, 15 pathways were significantly enriched ($P < 0.05$). The microbial-specific metabolic pathway was polycyclic aromatic hydrocarbon degradation, while the host was only significantly enriched in the steroid hormone biosynthesis pathway. In the host-microbe co-metabolism pathways, tyrosine metabolism, alanine, aspartate, and glutamate metabolism and others were significantly enriched (Fig. 6E). In summary, metabolic

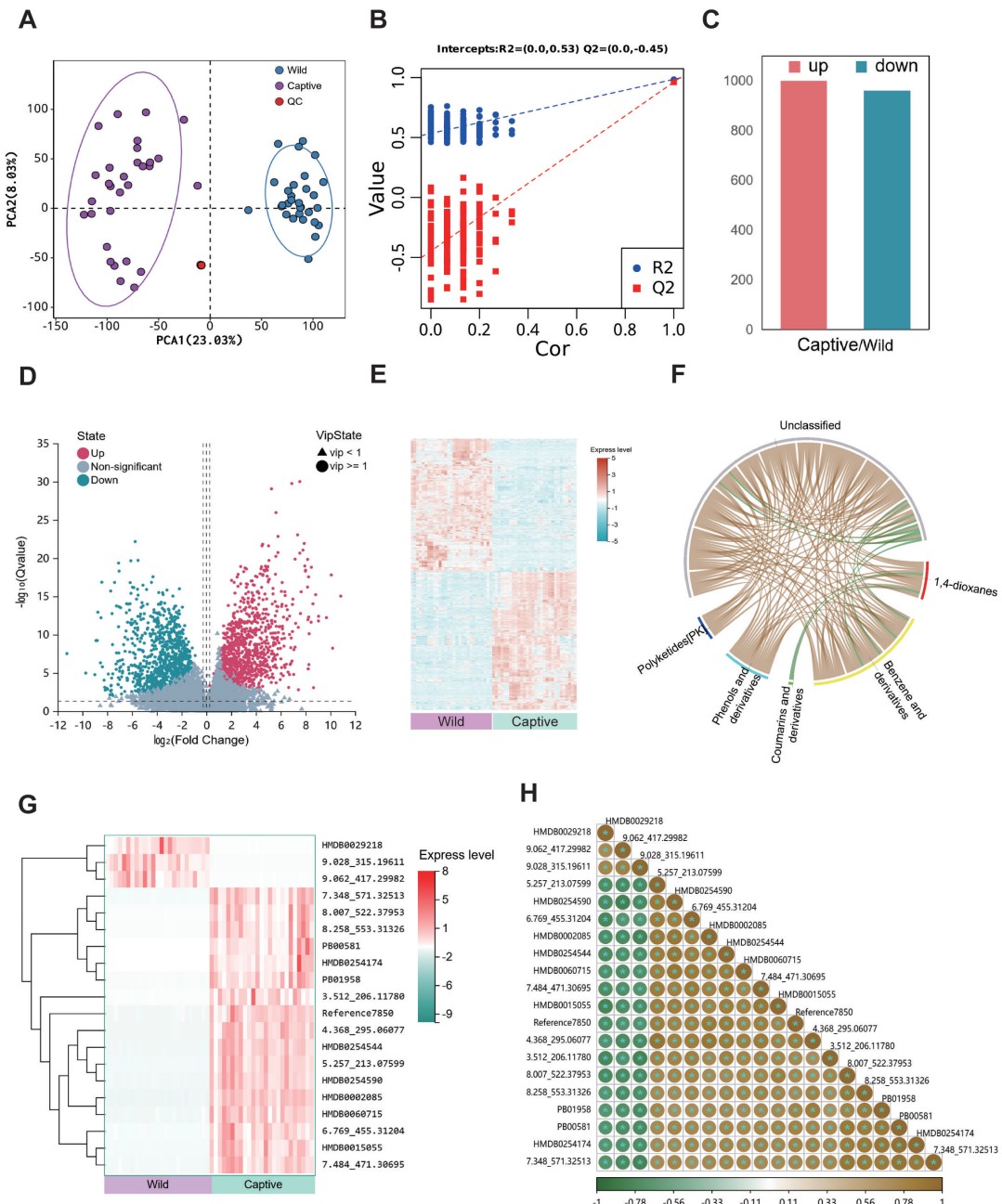

**FIG 5** Differential analysis of microbial metabolites in macaques. (A) PCA clustering plot. The *x*-axis represents the first principal component (PC1), and the *y*-axis represents the second principal component (PC2), with the numbers in parentheses indicating the scores of these components. In the PCA score plot, each point represents a sample, with purple indicating the wild group and green indicating the captive group. The ellipses denote the 95% confidence intervals. (B) Permutation test plot. The two points in the upper right corner represent the actual model's $R^2$ and $Q^2$ values. The points on the left indicate the results of the permutation test. (C) Bar chart of significant differential metabolites. *X*-axis: group comparisons; *y*-axis: number of differential metabolites. Red indicates significantly upregulated metabolites, while green indicates significantly downregulated metabolites. (D) Volcano plot of differential metabolites. The *x*-axis represents the log2-transformed fold change, and the *y*-axis represents the −log10-transformed q-value (*P*-value). Blue indicates significantly downregulated metabolites, red indicates significantly upregulated metabolites, and gray indicates non-significant metabolites. (E) Heatmap of differential metabolite abundance analysis. Each row in the heatmap represents a differential metabolite, with green to red corresponding to low to high expression levels. (F) Chord diagram of metabolite correlations. The starting points of the inner-circle links represent significant differential metabolites, and the arcs on the outer circle indicate the classification of these metabolites. Colored lines represent correlations within each metabolite class, with brown indicating positive correlations and green indicating negative correlations. (G) Heatmap of differential metabolite abundance analysis. Each row in the heatmap represents a differential metabolite, with green to red corresponding to low to high expression levels. (H) Heatmap of differential metabolite correlation analysis. Brown indicates strong positive

Fig 5 (Continued)

correlations, while green indicates strong negative correlations. Deeper colors represent higher absolute values of correlation coefficients between samples. Asterisks (*) denote *P*-values <0.05 for the correlation coefficients.

pathway analysis revealed captivity-induced perturbations in neuroactive and endocannabinoid pathways, with microbial-host co-metabolism driving 50 enriched pathways, including tyrosine metabolism, while the high-accuracy biomarker C10757 effectively discriminates between the wild and captive states.

## Stress environment alters the functional and metabolic pathways of the gut microbiota

### Integrative multi-omics comparative analysis

To explore the impact of psychological stress on the functional and metabolic pathways of the microbial community and its interactions with the host, we conducted an integrated analysis of metabolomics and metagenomics. Initially, we performed a combined analysis of microbial species and metabolite data from the wild and captive groups using PCA. The results showed a significant separation between the two groups, with samples within each group clustering closely together, indicating substantial differences in overall microbial community structure and metabolite profiles between the two groups (Fig. 7A and B). Correlation analysis of the microbiota at the genus level revealed that in the wild group, the dominant genera *Pasteurella* and *Succinivibrio* were associated with a larger number of significant metabolites. Specifically, *Pasteurella* was linked to 37 significant metabolites, while *Succinivibrio* was associated with 20. These genera were positively correlated with thiophene carboxylic acids and derivatives, para-dioxins, 1-benzopyrans, and oxanes ($P < 0.05$), and negatively correlated with 12-diazepines, lipoamides, and organic cyanides ($P < 0.05$). In contrast, in the captive group, the dominant genera *Ligilactobacillus*, *Weissella*, and *Enterobacter* were associated with a larger number of significant metabolites. These genera were positively correlated with dibenzoxazepines, phenylpiperidines, benzotriazoles, and quinolones and derivatives ($P < 0.05$), and negatively correlated with sulfones, alkanes, para-dioxins, and oxanes ($P < 0.05$, Fig. 7C).

### Functional multi-omics profiling

We conducted pathway enrichment analysis on functional genes and metabolites, and the results showed enrichment in pathways such as beta-alanine metabolism and alanine, aspartate, and glutamate metabolism, indicating that these pathways may play a key role in metabolic function and species distribution. Additionally, pathways such as biosynthesis of amino acids and ascorbate and aldarate metabolism were also highly enriched, suggesting that these pathways may have important regulatory roles in metabolic function and species distribution (Fig. 7D). Furthermore, we found that in the wild group, metabolites such as amino acids, peptides and their analogs, coumarins and derivatives, and non-metal phosphates were more abundant. Among these metabolites, *Petrimonas, Arenibacter, Butyricimonas, Weissella*, and *Odoribacter* were positively correlated with coumarins and derivatives ($P < 0.05$). Based on the correlation analysis of differential genera and metabolites in the wild and captive Hainan macaques, *Weissella* showed a highly significant positive correlation with coumarins and derivatives ($P < 0.01$, $r > 0.3$), and *Odoribacter* showed a significant positive correlation ($P < 0.05$, $r > 0.2$). *Lentibacillus* and *Marivirga* were negatively correlated with coumarins and derivatives ($P < 0.05$). In the captive group, metabolites such as fatty acyls, organic acids, phloroglucinols, and carbonyl compounds were more abundant. *Candidatus_Sulcia, Gilliamella*, and *Candida* were positively correlated with these metabolites ($P < 0.05$), while *Thermogutta* was negatively correlated ($P < 0.05$, Fig. 7E). Integrated multi-omics analysis revealed stress-induced host-microbe metabolic rewiring, with wild macaques showing amino acid/coumarin-associated microbiota and captive specimens exhibiting fatty acid/

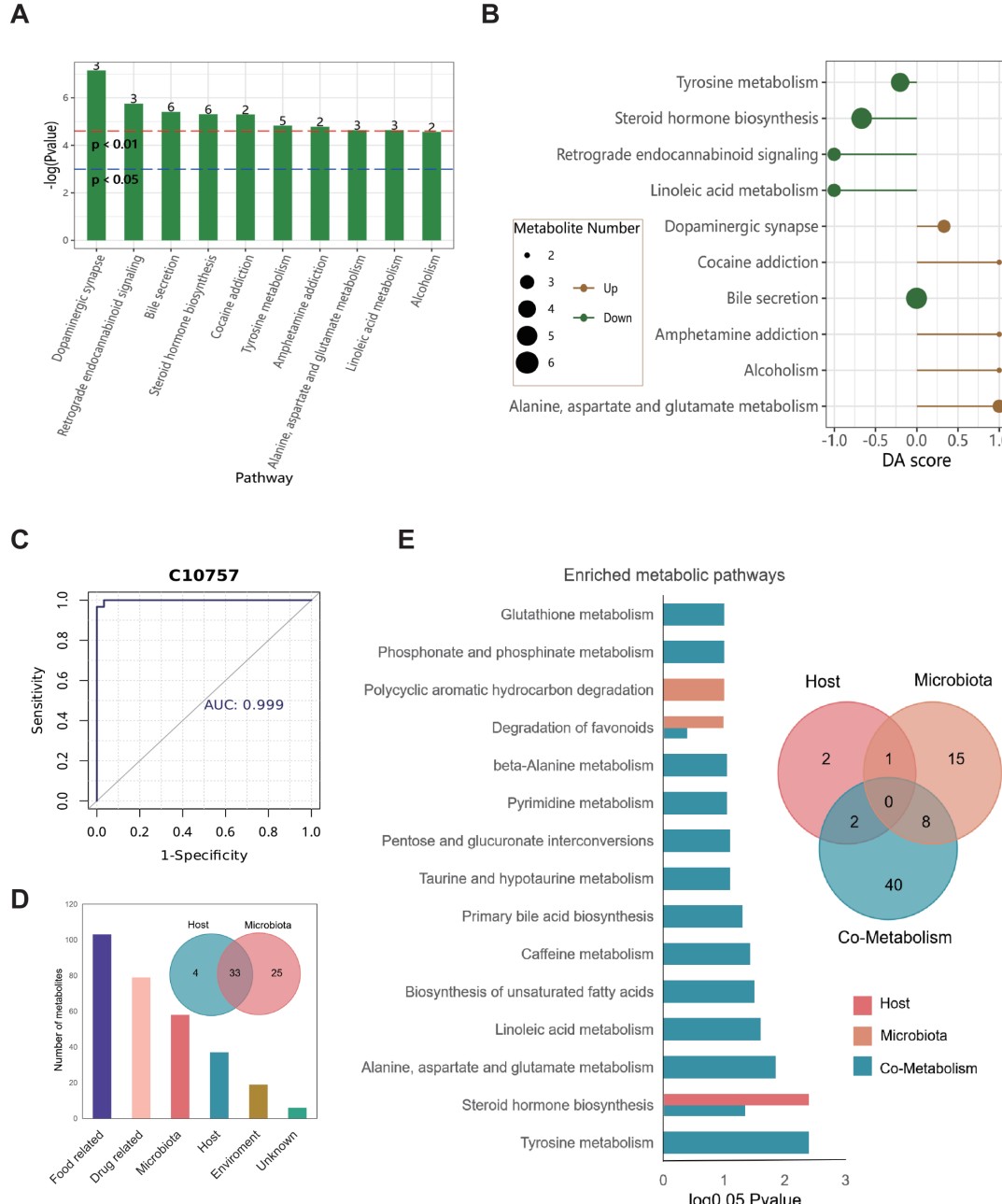

**FIG 6** Functional enrichment and source tracking of differential metabolites. (A) Bar chart of enrichment analysis. Metabolic pathways with −log($P$-value) above the red auxiliary line have $P$-values less than 0.01, while those above the blue auxiliary line have $P$-values less than 0.05. The numbers on the bars represent the number of differential metabolites in the corresponding pathways. (B) Pathway enrichment score plot. The $y$-axis indicates the names of metabolic pathways, and the $x$-axis represents the differential abundance score (DA score). A score of 1 indicates that all annotated differential metabolites in the pathway are upregulated, while a score of −1 indicates they are downregulated. The length of the line segments represents the absolute value of the DA score. (C) ROC curve. The $x$-axis represents 1-specificity, and the $y$-axis represents sensitivity. The area under the curve is the AUC value. (D) Bar chart and Venn diagram of metabolite counts by source. The $x$-axis indicates the source of metabolites, and the $y$-axis represents the number of metabolites. (E) Bar chart and Venn diagram of pathway enrichment for metabolites from the same source (only statistically significant pathways are shown).

organic acid-dominant profiles linked to distinct genera, demonstrating microbiome-mediated metabolic adaptation to psychological stress.

In summary, we elucidate the mechanisms (Fig. 8) that psychological stress alters gut microbiota composition and function in Hainan macaques, particularly affecting neuroactive coumarin metabolites. We identified metabolite C10757 as a potential stress

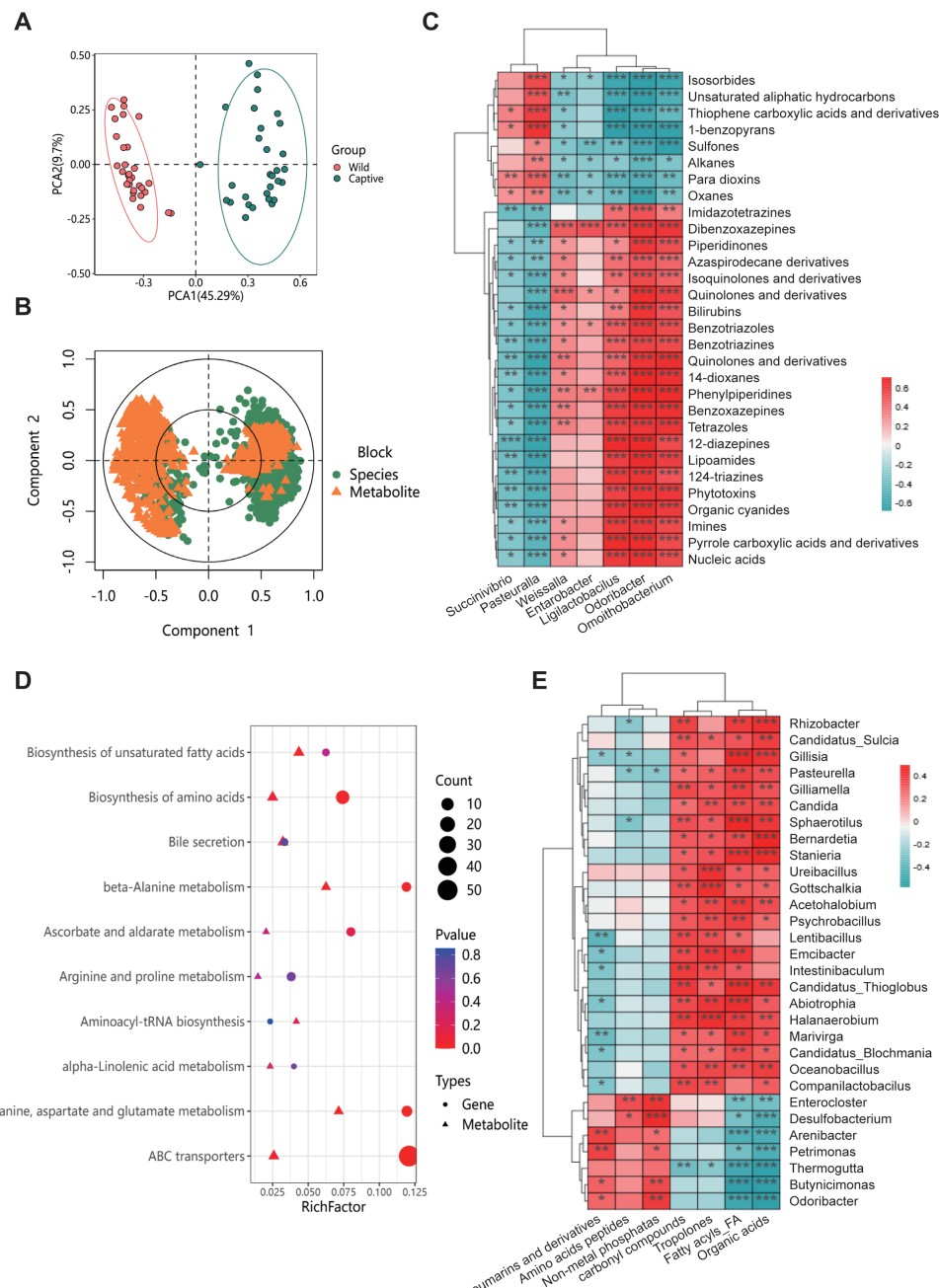

**FIG 7** Integrated analysis of metabolomics and metagenomics in macaque microbiota. (A) Combined PCA clustering plot of species and metabolite data. (B) Canonical correspondence analysis plot. The inner circle represents a correlation coefficient of 0.5, and the outer circle represents a correlation coefficient of 1. Species are indicated by green dots, and metabolites are indicated by orange triangles. (C) Heatmap of correlation analysis between microbiota and differential metabolites. (D) Enrichment analysis of the plot of functional genes and metabolite pathways. The x-axis represents the enrichment factor (RichFactor). Triangles denote functional gene pathways, and circles denote metabolic pathways. (E) Heatmap of correlation analysis between the metabolome and differential microbiota.

biomarker, suggesting microbiome-mediated neurological modulation. These findings support microbiota-targeted interventions for stress management and highlight gut microbiome monitoring in wildlife conservation.

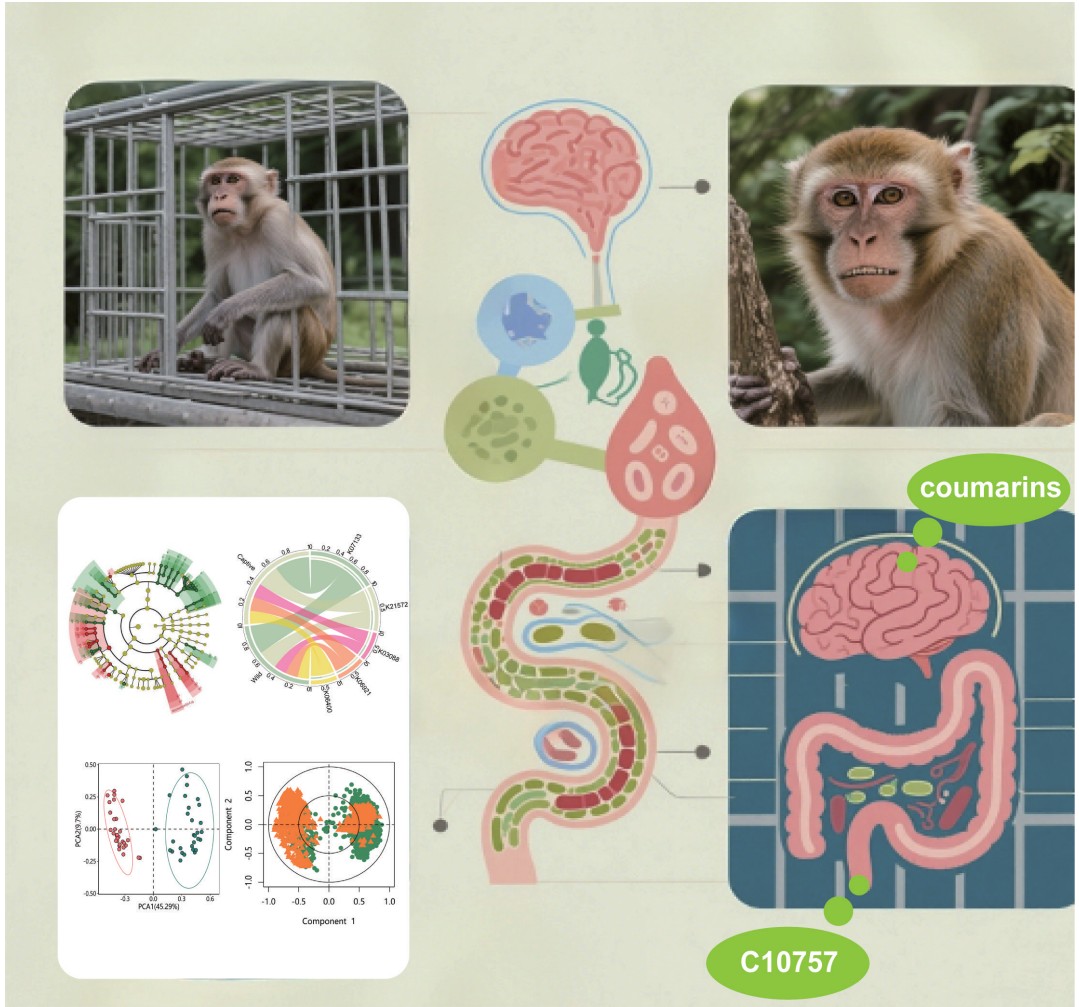

**FIG 8** Psychological stress alters gut microbiota and neuroactive metabolites in macaques, suggesting microbiome-mediated stress biomarkers for conservation.

## DISCUSSION

Psychological stress has long been recognized as a significant factor influencing the homeostasis of gut microbiota (25, 26). Recent studies have shown that stress can induce alterations in microbial composition and function, which in turn can affect host health (27, 28). In this study, we observed that stress-induced changes in the gut microbiota were closely associated with shifts in the production of key metabolites, particularly coumarins. These metabolites are known to interact with the nervous system, suggesting a potential feedback loop where stress alters the gut microbiota, which then modulates neurological function through the production of bioactive compounds. This finding underscores the importance of considering the gut-brain axis in understanding the broader impacts of psychological stress on health. Specifically, the observed changes in the gut microbiota and associated metabolites may provide a mechanistic link between stress and neurological disorders, highlighting the potential for microbiota-based interventions to mitigate stress-related health issues.

Our comprehensive analysis identified several key metabolites that are differentially expressed under stress conditions. These metabolites, including specific coumarin derivatives, can serve as biomarkers to detect and quantify the impact of environmental stressors on psychological well-being (29). By leveraging the unique metabolic profiles of the gut microbiota, we can develop non-invasive methods to assess the extent of

stress-induced health risks (30, 31). This approach not only enhances our ability to monitor stress in wildlife populations but also provides a valuable tool for assessing the mental health of animals in captive environments (32, 33). The identification of these biomarkers represents a significant step forward in the field of stress detection, offering a more precise and objective means of evaluating the effects of stress on health. Furthermore, the use of these metabolites as biomarkers could facilitate early intervention and management strategies to mitigate the adverse effects of stress on both the wild and captive animals (34, 35).

This study reveals critical insights into how psychological stress disrupts gut microbiota in wild animals, advancing our understanding of the gut-brain axis in non-captive species. Our findings demonstrate that gut microbial profiles serve as sensitive, non-invasive biomarkers for stress, offering conservationists a practical tool to monitor wildlife health and detect early signs of environmental stressors. The results also underscore the ethical imperative to integrate psychological stress assessments (e.g., microbiota profiling coupled with cortisol monitoring) into research protocols, ensuring both animal welfare and data reliability in wild animal studies. Mechanistically, the observed microbial and metabolic shifts (e.g., reduced *Lactobacillus* and short chain fatty acid depletion) align with findings in captive species, suggesting evolutionary conservation of gut-brain pathways and potential for cross-species therapeutic strategies. Moving forward, longitudinal studies should explore microbiota-targeted interventions to enhance stress resilience in wild populations, while comparative research across species will clarify the generality of these effects. By linking psychobiological mechanisms to conservation practice and ethical research design, this work provides a foundational framework to improve wildlife management strategies and refine experimental approaches, ultimately promoting both ecological health and humane scientific inquiry.

## MATERIALS AND METHODS

### Sample collection and sources

Wild macaques were studied at Nanwan Monkey Island in the Lingshui Macaque Nature Reserve, Hainan Province, where 100 fresh fecal samples Ling Shui (LS) were collected from 30 individuals. In consideration of wildlife welfare and conservation ethics, captive rhesus macaques were selected from a wildlife rescue center, matching the wild macaques in age, weight, health status, and diet (plants, fruits, etc.), collecting 90 fresh fecal samples BaoTing (BT) from 30 individuals. Each sample was collected within 2 min of defecation, with the uncontaminated internal portion placed into sterile cryovials, then preserved in liquid nitrogen and transported to the laboratory for storage at −80℃. Throughout the sampling process, no methods potentially harmful to the macaques were employed, and the study complied with relevant laws and regulations, including the Wildlife Protection Law and the Nature Reserve Management Regulations.

### Analytical software and databases

Data filtering was performed using SOAPnuke (v1.5.0), data alignment with Bowtie2 (2.2.5), and data processing with Samtools (1.2). Metagenomic assembly was conducted using MEGAHIT, functional annotation with DIAMOND, and species annotation with Kraken2. MetOrigin platform (http://metorigin.met-bioinformatics.cn) was utilized for metabolite tracing analysis. Data analysis and visualization were carried out using R (4.1.2), with reference databases including KEGG, HMDB, CARD, CAZy, Compound Discoverer v.3.3, mzCloud, ChemSpider, and BioSecure Multimodal Database (BMDB).

### Metagenomic sample processing

The central portion of fecal samples, uncontaminated by the external environment, was extracted and stored at −80℃. Genomic DNA was extracted following the MagPure

Stool DNA KF Kit B (MFDP-0101) protocol, and qualified samples were used for library construction. The DNBSEQ platform was employed for metagenomic analysis of macaque fecal microbiota.

Library construction workflow: (i) DNA concentration and integrity were assessed, with qualified samples proceeding with library preparation. (ii) A Covaris sonicator was used to fragment a specified amount of metagenomic DNA. (iii) To focus sample bands around 200 bp–400 bp, magnetic bead-based fragment selection was performed on the fragmented samples. (iv) A reaction system was configured to repair double-stranded cDNA ends and append an A base to the 3´ end, followed by adapter ligation to the DNA. (v) A PCR reaction system was set up and programmed to amplify the ligation products, with magnetic beads used for product purification and recovery. (vi) PCR products were denatured into single strands, and a circularization reaction system was configured and reacted at an appropriate temperature to obtain single-stranded circular products. Linear DNA molecules were not circularized, yielding the final library. (vii) Pre-sequencing concentration detection was performed on the circularized products. (viii) DNBSEQ0T7, PE150 sequencing was conducted. DNA extraction and sequencing were completed on the BGI Genomics proprietary sequencing platform.

## Metagenomic data analysis

Raw sequencing data underwent filtering to obtain clean data. MEGAHIT software was used to assemble the metagenome from the quality-controlled clean data. Metagenomic genes were predicted *de novo* using Meta Gene Mark, and redundant gene prediction results from each sample were processed using CD-HIT software. Gene expression levels were quantified using Salmon software, with TPM values representing normalized gene abundance.

Non-redundant genes were functionally annotated using DIAMOND (BLASTP), with species profiling performed via Kraken2/Bracken. Functional analyses employed KEGG (metabolic pathways), CAZy (carbohydrate-active enzymes), and CARD (antibiotic resistance genes). Alpha (Chao1/Shannon/Simpson) and beta (Bray-Curtis/Jensen-Shannon) diversity metrics were computed in R, while differential abundance was assessed using DESeq2 and LEfSe permutational multivariate analysis of variance (PERMANOVA for PCoA validation). Pathway enrichment (reporter scores) and CAZy/antibiotic resistance genes (ARG) comparisons (STAMP; Wilcoxon/Kruskal-Wallis, $P < 0.05$) were visualized via extended bar charts and boxplots.

## Metabolomics sample preparation and detection

Corresponding samples for metabolomics were thawed at 4℃, then processed through grinding, ultrasonication in a 4℃ water bath, static placement at −20℃, centrifugation, and resuspension. An appropriate amount of supernatant was taken as a quality control sample to assess the repeatability and stability of liquid chromatograph mass spectrometer (LC-MS) analysis.

Metabolites were separated and detected using a Waters UPLC I-Class Plus (Waters, Milford, MA) coupled with a Q Exactive high-resolution mass spectrometer (Thermo Fisher Scientific, Bremen, Germany). Chromatographic conditions were as follows: for positive ion mode, mobile phase A (0.1% formic acid in water) and mobile phase B (0.1% formic acid in methanol); for negative ion mode, mobile phase A (10 mM ammonium formate in water) and mobile phase B (10 mM ammonium formate in 95% methanol). Solvents were eluted in a gradient, with a flow rate of 0.35 mL/min, column temperature at 45℃, and injection volume of 5 µL. Mass spectrometry conditions included a scanning mass-to-charge ratio range of 70–1,050, with the top 3 precursor ions selected for fragmentation at energies of 20 eV, 40 eV, and 60 eV. Ion source parameters were as follows: sheath gas flow rate of 40, auxiliary gas flow rate of 10, spray voltage of 3.80 in positive ion mode and 3.20 in negative ion mode, ion transfer tube temperature of 320℃, and auxiliary gas heater temperature of 350℃.

## Metabolomics data analysis

Raw mass spectrometry data were imported into Compound Discoverer 3.3 (Thermo Fisher Scientific, USA) software. The exported results were further processed in metaX for data preprocessing and subsequent analysis. Preprocessing included probabilistic quotient normalization to obtain relative peak areas and batch effect correction using locally weighted polynomial regression fitting QC-based Robust LOESS Signal Correction (QC-RLSC). Compounds with relative peak area coefficients of variation greater than 30% in QC samples were removed.

Principal component analysis was performed in R to reduce data dimensions and ensure data quality. Metabolites were classified and annotated using the HMDB and KEGG databases. Variable importance in projection (VIP) values were calculated to assist in the selection of metabolic biomarkers. Data were log2-transformed to establish partial least squares discriminant analysis (PLS-DA) models for intergroup comparison and differential metabolite screening. Differential metabolites were identified using VIP values >1 in combination with univariate statistical analysis ($P < 0.05$). Results were visualized using R. Differential metabolites were traced using the MetOrigin platform. Spearman and Pearson correlation analyses were conducted in R to examine the relationships between differential microbes and metabolites in Hainan macaque feces.

## Statistical analysis

Differences were considered statistically significant at $P < 0.05$ and highly significant at $P < 0.01$. Spearman and Pearson correlation analyses were performed in R to analyze the relationships between differential microbes and metabolites in Hainan macaque feces, with results visualized in plots.

### ACKNOWLEDGMENTS

This work was supported by the Technical Innovation Project of Research Institutions in Hainan Province (SQKY2022-0016) and the Research Startup Fund Project for Introduced Talent at the Hainan Academy of Agricultural Sciences (HNXM2024RCQD06).

### AUTHOR AFFILIATIONS

[1]Innovation Center of Academician Xia Xianzhu's Team, Key Laboratory of Tropical Animal Breeding and Disease Research, Institute of Animal Science and Veterinary Medicine, Hainan Academy of Agricultural Sciences, Haikou, China

[2]State Key Laboratory of Stem Cell and Reproductive Biology, Institute of Zoology, Stem Cell and Regenerative Medicine Innovation Institute, Chinese Academy of Sciences, Beijing, China

[3]Sanya Institute, Hainan Academy of Agricultural Sciences, Sanya, China

[4]College of Veterinary Medicine, South China Agricultural University, Guangzhou, China

[5]Tropical Crop Genetic Resource Research Institute, Chinese Academy of Tropical Agricultural Sciences, Haikou, China

[6]State Key Laboratory of Animal Biotech Breeding, Institute of Animal Science, Chinese Academy of Agricultural Sciences (CAAS), Beijing, China

### AUTHOR ORCIDs

Yanfang Wang ⓘ http://orcid.org/0000-0003-2778-0759
Jianguo Zhao ⓘ http://orcid.org/0000-0001-6587-4823
Jingli Yuan ⓘ http://orcid.org/0009-0002-3502-1672

## FUNDING

| Funder | Grant(s) | Author(s) |
|---|---|---|
| Technical Innovation Project of Research Institutions in Hainan Province | SQKY2022-0016 | Jingli Yuan |
| Research startup Fund Project for Introduced Talent at the Hainan Academy of Agricultural Sciences | HNXM2024RCQD06 | Jingli Yuan |

## AUTHOR CONTRIBUTIONS

Zewen Sun, Conceptualization, Formal analysis, Investigation, Resources, Writing – original draft, Writing – review and editing | Jun Wang, Data curation, Formal analysis | Ruiping Sun, Formal analysis, Methodology | Baozhen Liu, Resources, Visualization | Keqi Cai, Formal analysis, Methodology | Xinyuan Zhao, Investigation, Methodology | Yanfang Wang, Conceptualization, Writing – review and editing | Jianguo Zhao, Conceptualization, Funding acquisition, Resources, Writing – review and editing | Jingli Yuan, Conceptualization, Funding acquisition, Investigation, Supervision, Writing – original draft, Writing – review and editing

## DATA AVAILABILITY

All data are available in the main article or the supplementary materials and from the corresponding author upon reasonable request. The sequencing data from this study has been deposited in the CNSA database (https://db.cngb.org/cnsa/) under accession number CNP0007258.

## ADDITIONAL FILES

The following material is available online.

### Supplemental Material

**Supplemental material (Spectrum01338-25-S0001.docx).** Fig. S1 and S2; Tables S1 and S2.

### Open Peer Review

**PEER REVIEW HISTORY (review-history.pdf).** An accounting of the reviewer comments and feedback.

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
