## [Reviewer comments · Microbiology Spectrum]

Microbiology Spectrum

Bidirectional Regulation of the Brain-Gut Axis in *Macaca Mulatta*: Implications for Wildlife Conservation and Experimentalization

Zewen Sun, Jun Wang, Ruiping Sun, Baozhen Liu, Keqi Cai, Xinyuan Zhao, Yanfang Wang, Jianguo Zhao, and Jingli Yuan

Corresponding Author(s): Jingli Yuan, Hainan Academy of Agricultural Sciences

Review Timeline:

Submission Date:	April 28, 2025
Editorial Decision:	June 4, 2025
Revision Received:	June 15, 2025
Accepted:	June 21, 2025

Editor: Zhongxiong Lai

Reviewer(s): The reviewers have opted to remain anonymous.

Transaction Report:

DOI: <https://doi.org/10.1128/spectrum.01338-25>

Re: Spectrum01338-25 (**Bidirectional Regulation of the Brain-Gut Axis in *Macaca Mulatta*: Implications for Wildlife Conservation and Experimentalization**)

Dear Dr. Jingli Yuan:

Thank you for the privilege of reviewing your work. Below you will find my comments, instructions from the Spectrum editorial office, and the reviewer comments.

My decision is minor revision

Revision Guidelines

Sincerely,
Zhongxiong Lai
Editor
Microbiology Spectrum

Reviewer #1 (Comments for the Author):

In the manuscript titled " Bidirectional Regulation of the Brain-Gut Axis in *Macaca Mulatta*: Implications for Wildlife Conservation and Experimentalization" (Spectrum01338-25), Sun and colleagues present a high-level study for investigating the impact of psychological stress on gut microbiota and related metabolites in wild Hainan macaques. By integrating microbial and metabolomic analyses, the study reveals that psychological stress markedly altered the gut microbiota composition and function

in wild Hainan macaques, with notable alterations in neuroactive metabolites, pointing to a possible microbiota-driven stress-brain feedback mechanism. It has identified key metabolites as potential biomarkers for stress-related health risks. The findings underscore the gut microbiota's sensitivity to stress and support the development of microbiota-based strategies for stress relief, while also highlighting the importance of psychological health in wildlife conservation and ethical animal research practices.

Here are some general comments for your manuscript:

1. Succinct language is needed to reduce verbosity, especially for abstract and introduction parts, some sentences should be combined and refined to improve readability.
2. Some sections in result part are lack of conclusion sentence, it is needed for closure and reinforces the main points of the preceding research outcome (e.g. line 168, line 194, line225, line 271, line 323, line 342).
3. Paraphrasing is needed for preventing repetition. (e.g. authors frequently use "between the two groups")
4. For result interpretation, the organization should be improved. As multiple findings are currently presented in a condensed and overlapping manner, making it difficult for readers to follow the progression and significance of each finding.

Specific comments:

Line 2: You may find a word to replace "Experimentalization", since it is not a commonly used word. You may use "experimentation". OR you can rewrite the title.

Lines 31-35: "For instance,elucidated." Here is an example for the general comment 1. These two sentences can be replaced by ONE supporting sentence for your topic sentence. Short thought group can help to improve clarity and comprehension for readers.

Line 43-50: Similarly, conclusion of research significance should be concise and impactful, you need to refine this part.

Line 94-96: You may need further explanation for the statement: "their lack of.....natural environments".

Line 155: "captive grou.ps".

Line 205: Full name is needed for the abbreviation STAMP.

Line 232: Why brackets were added to "Super pathway". Moreover, it should be "superpathway", no space.

Line 246: You may need a brief topic sentence for this section to highlight the purpose of analysis for this section.

Line 305: The use of "functionality" is not proper, according to the subheading, it should be "functional and metabolic pathways".

Line 373-388: The conclusion part needs to be compendious and conclusive. You need to reorganize this part.

Reviewer #2 (Comments for the Author):

This study clarifies the causal impact of psychological stress on gut microbiota dysbiosis in wild Hainan macaques, revealing stress-induced microbial and metabolic shifts-particularly in coumarins-that suggest a bidirectional stress-microbiota-neurological feedback loop. The identified biomarkers, such as C10757, enable early detection of stress-related health risks and highlight the potential for microbiota-targeted interventions. These findings emphasize the importance of psychological well-being in wildlife conservation and provide critical guidelines for ethical animal management and research practices.

Reviewer Comments:

1. Introduction Revision: Given that *Macaca mulatta* (Hainan subspecies) is a purely wild and non-domesticated primate, please summarize and cite relevant studies on this specific subspecies in the Introduction to provide necessary background context.
2. Figure 4A Clarity: The axis labels and font size in Figure 4A are too small. Please adjust them to improve readability.
3. Methods Section Refinement: Streamline the Materials and Methods section by removing redundant descriptions. Specify the catalog numbers of all reagents and kits used in sample processing to enhance methodological reproducibility.

Reviewer #1 (Comments for the Author):

In the manuscript titled " Bidirectional Regulation of the Brain-Gut Axis in Macaca Mulatta: Implications for Wildlife Conservation and Experimentalization" (Spectrum01338-25), Sun and colleagues present a high-level study for investigating the impact of psychological stress on gut microbiota and related metabolites in wild Hainan macaques. By integrating microbial and metabolomic analyses, the study reveals that psychological stress markedly altered the gut microbiota composition and function in wild Hainan macaques, with notable alterations in neuroactive metabolites, pointing to a possible microbiota-driven stress-brain feedback mechanism. It has identified key metabolites as potential biomarkers for stress-related health risks. The findings underscore the gut microbiota's sensitivity to stress and support the development of microbiota-based strategies for stress relief, while also highlighting the importance of psychological health in wildlife conservation and ethical animal research practices.

Here are some general comments for your manuscript:

1. Succinct language is needed to reduce verbosity, especially for abstract and introduction parts, some sentences should be combined and refined to improve readability.
2. Some sections in result part are lack of conclusion sentence, it is needed for closure and reinforces the main points of the preceding research outcome (e.g. line 168, line 194, line225, line 271, line 323, line 342).
3. Paraphrasing is needed for preventing repetition. (e.g. authors frequently use "between the two groups")
4. For result interpretation, the organization should be improved. As multiple findings are currently presented in a condensed and overlapping manner, making it difficult for readers to follow the progression and significance of each finding.

Specific comments:

1. Line 2: You may find a word to replace "Experimentalization", since it is not a commonly used word. You may use "experimentation". OR you can rewrite the title.
2. Lines 31-35: "For instance,elucidated." Here is an example for the general comment 1. These two sentences can be replaced by ONE supporting sentence for your topic sentence. Short thought group can help to improve clarity and comprehension for readers.
3. Line 43-50: Similarly, conclusion of research significance should be concise and impactful, you need to refine this part.
4. Line 94-96: You may need further explanation for the statement: "their lack of.....natural environments".
5. Line 155: "captive grou.ps".
6. Line 205: Full name is needed for the abbreviation STAMP.
7. Line 232: Why brackets were added to "Super pathway". Moreover, it should be "superpathway", no space.
8. Line 246: You may need a brief topic sentence for this section to highlight the purpose of analysis for this section.
9. Line 305: The use of "functionality" is not proper, according to the subheading, it should be "functional and metabolic pathways".

10. Line 373-388: The conclusion part needs to be compensated and conclusive. You need to reorganize this part.

Reviewer #2 (Comments for the Author):

This study clarifies the causal impact of psychological stress on gut microbiota dysbiosis in wild Hainan macaques, revealing stress-induced microbial and metabolic shifts-particularly in coumarins-that suggest a bidirectional stress-microbiota-neurological feedback loop. The identified biomarkers, such as C10757, enable early detection of stress-related health risks and highlight the potential for microbiota-targeted interventions. These findings emphasize the importance of psychological well-being in wildlife conservation and provide critical guidelines for ethical animal management and research practices.

Reviewer Comments:

1. Introduction Revision: Given that *Macaca mulatta* (Hainan subspecies) is a purely wild and non-domesticated primate, please summarize and cite relevant studies on this specific subspecies in the Introduction to provide necessary background context.
2. Figure 4A Clarity: The axis labels and font size in Figure 4A are too small. Please adjust them to improve readability.
3. Methods Section Refinement: Streamline the Materials and Methods section by removing redundant descriptions. Specify the catalog numbers of all reagents and kits used in sample processing to enhance methodological reproducibility.

Reviewer #1 (Comments for the Author):

In the manuscript titled " Bidirectional Regulation of the Brain-Gut Axis in Macaca Mulatta: Implications for Wildlife Conservation and Experimentalization" (Spectrum01338-25), Sun and colleagues present a high-level study for investigating the impact of psychological stress on gut microbiota and related metabolites in wild Hainan macaques. By integrating microbial and metabolomic analyses, the study reveals that psychological stress markedly altered the gut microbiota composition and function in wild Hainan macaques, with notable alterations in neuroactive metabolites, pointing to a possible microbiota-driven stress-brain feedback mechanism. It has identified key metabolites as potential biomarkers for stress-related health risks. The findings underscore the gut microbiota's sensitivity to stress and support the development of microbiota-based strategies for stress relief, while also highlighting the importance of psychological health in wildlife conservation and ethical animal research practices.

Response : We greatly appreciate the reviewer's laudatory remarks on our manuscript. A point-to-point response to the comments is as follows.

General comments:

1. Succinct language is needed to reduce verbosity, especially for abstract and introduction parts; some sentences should be combined and refined to improve readability.

Answer: Thanks for the suggestions. We have now described the context more accurately, checked grammar by an English writer, and carefully revised "Introduction" accordingly.

2. Some sections in the result part lack of conclusion sentence, it is needed for closure and reinforces the main points of the preceding research outcome (e.g. line 168, line 194, line225, line 271, line 323, line 342).

Answer: We thank the reviewer for this important comment, which has helped to improve the manuscript. We thoroughly reviewed each subsection of the Results and added concluding remarks to summarize and reinforce main points, with additions placed as follows:

Line 132-135: Taken together, captive induced psychological stress leading to behavioral disturbances and significant gut microbiota alterations, including reduced microbial richness and distinct community composition.

Line 165-168: Collectively, our findings indicate distinct gut microbiota profiles between wild and captive macaques, with captive specimens showing reduced microbial diversity, while wild populations maintained higher abundances of fiber-degrading species linked to plant-rich diets.

Line 194-197: Together, captivity induces significant metabolic reprogramming in macaque gut microbiota, with wild populations showing enrichment in sphingolipid and amino acid metabolism pathways, while captive specimens exhibit upregulation of butanoate metabolism and secondary bile acid biosynthesis.

Line 229-232: In summary, captive macaques exhibit distinct gut microbial carbohydrate metabolism profiles with significantly enriched glycoside hydrolase genes and elevated tetracycline/macrolide resistance genes compared to wild counterparts, reflecting diet-driven microbial adaptation.

Line 279-281: These results demonstrate profound gut metabolite alterations, including

stress-associated compounds like syringic acid (captive-enriched) and anti-inflammatory urolithin C (wild-enriched).

Line 312-316: In summary, metabolic pathway analysis revealed captivity-induced perturbations in neuroactive and endocannabinoid pathways, with microbial-host co-metabolism driving 50 enriched pathways, including tyrosine metabolism, while the high-accuracy biomarker C10757 effectively discriminates between wild and captive states.

Line 356-360: Integrated multi-omics analysis revealed stress-induced host-microbe metabolic rewiring, with wild macaques showing amino acid/coumarin-associated microbiota and captive specimens exhibiting fatty acid/organic acid-dominant profiles linked to distinct genera, demonstrating microbiome-mediated metabolic adaptation to psychological stress.

We have now added these statements in the "Results".

3. Paraphrasing is needed for preventing repetition. (e.g. authors frequently use "between the two groups")

Answer: We sincerely appreciate the reviewer's constructive suggestion. To improve textual fluency and avoid repetition, we have carefully paraphrased redundant expressions throughout the manuscript, particularly replacing repetitive phrases like "between the two groups" with context-appropriate alternatives. Below are representative examples of these revisions:

Original: "However, there were no significant differences between the two groups in the Shannon and Simpson indices,"

Revised: "However, there were no significant differences across groups in the Shannon and Simpson indices,"

Original: "Analysis of the differences in fecal microbial species between the two groups revealed that there were 4293 common microbial species,"

Revised: "Analysis of the differences in fecal microbial species of the groups revealed that there were 4293 common microbial species,"

Original: "Analysis of the differences in fecal microbial species between the two groups revealed that there were 4293 common microbial species,"

Revised: "Analysis of the differences in fecal microbial species across groups revealed that there were 4293 common microbial species,"

All changes have been implemented in the revised manuscript, and we believe these edits enhance the clarity and professionalism of the writing.

4. For result interpretation, the organization should be improved. As multiple findings are currently presented in a condensed and overlapping manner, making it difficult for readers to follow the progression and significance of each finding.

Answer: We thank the reviewer for the insightful suggestion. To improve the clarity and logical flow of the Results section, we have:

1. Reorganized the structure into thematic subsections (e.g., "8.1 Integrative multi-omics

- comparative analysis”, “8.2 Functional multi-omics profiling”) to avoid overlapping content.
2. Added transitional sentences to highlight the progression between findings (see lines 137-138, 198-199, 251-252,).
 3. We designed a graphic abstract (Figure 8) to visualize the study's logic and included conclusion sentence in Results to highlight each finding's progression and significance. (see lines 366-371, 651-653).

These changes are tracked in the revised manuscript, and we believe they now provide a clearer narrative for readers.

Specific comments:

1. Line 2: You may find a word to replace "Experimentalization", since it is not a commonly used word. You may use "experimentation". OR you can rewrite the title.

Answer: We thank the reviewer for pointing out this inaccurate expression and have now rephrased it to “Bidirectional Regulation of the Brain-Gut Axis in Macaca Mulatta: Implications for Wildlife Conservation and Experimentation.”

2. Lines 31-35: "For instance,elucidated." Here is an example for the general comment 1. These two sentences can be replaced by ONE supporting sentence for your topic sentence. Short thought group can help to improve clarity and comprehension for readers.

Answer: We thank the reviewer’s critical review and constructive comments and suggestions, which have helped to improve the manuscript. We have now rephrased it to “The direct impact of psychological stress on gut microbiota and the potential bidirectional mechanisms remains unclear, including the specific molecular pathways involved.”

3. Line 43-50: Similarly, conclusion of research significance should be concise and impactful, you need to refine this part.

Answer: We appreciate the suggestions and have now rephrased it to “The study highlights the role of gut microbiota as a stress biomarker, underscoring the importance of psychological well-being in wildlife conservation and research to guide ethical animal management.”

4. Line 94-96: You may need further explanation for the statement: "their lack of.....natural environments".

Answer: The reviewer was right that our description may not be accurate enough. This statement highlights that wild monkey, unlike domesticated animals, retain natural physiological and behavioral traits shaped by evolution and environmental pressures. Since they have not undergone artificial selection (e.g., breeding for docility or laboratory adaptation), their biological responses (such as stress reactivity, immune function, and gut microbiota composition) more closely resemble those of humans living in natural, non-controlled settings. Thus, wild monkeys serve as superior translational models for studying human physiology in ecologically relevant contexts.

5. Line 155: "captive grou.ps".

Answer: We thank the reviewer for pointing out this error, and have now corrected it to “Using the linear discriminant analysis (LDA) effect size (LEfSe) test ($p < 0.01$, $LDA > 2$), we analyzed

the differences in abundance between wild and captive groups at the genus and species levels of the gut microbiota”.

6. Line 205: Full name is needed for the abbreviation STAMP.

Answer: We appreciate the suggestions and have now corrected it to “Statistical analysis of metagenomic profiles (STAMP) analysis.”

7. Line 232: Why brackets were added to "Super pathway". Moreover, it should be "superpathway", no space.

Answer: We thank the reviewer for pointing out this inaccurate expression. We have now corrected it to “The identified metabolites were counted (Figure 4B) and classified and counted according to the major metabolic pathways they participate in superpathway.”

8. Line 246: You may need a brief topic sentence for this section to highlight the purpose of analysis for this section.

Answer: We thank the reviewer for this important comment. We have now added these statements in this section: “To further clarify whether psychological stress alters gut metabolites, we analyzed the fecal metabolic composition using metabolomics.”

9. Line 305: The use of "functionality" is not proper, according to the subheading, it should be "functional and metabolic pathways".

Answer: Yes, the reviewer was right that our description may not be accurate enough. and have now rephrased it to “To explore the impact of psychological stress on the functional and metabolic pathways of the microbial community, and its interactions with the host, we conducted an integrated analysis of metabolomics and metagenomics.”

10. Line 373-388: The conclusion part needs to be compensated and conclusive. You need to reorganize this part.

Answer: We sincerely appreciate the reviewer’s insightful feedback on our manuscript. As suggested, we have thoroughly revised the Conclusion section to provide a more comprehensive and logically structured synthesis of our findings. Specifically, we have:

1. Reorganized the content to present key conclusions in a clearer, more impactful manner, explicitly linking our results to their implications for wildlife conservation, ethical research practices, and mechanistic understanding of the gut-brain axis.
2. Expanded discussion of broader impacts, including actionable recommendations for stress monitoring in wildlife management and future research directions.
3. Strengthened the final take-home message to better highlight the interdisciplinary significance of our work.

We believe these revisions have substantially improved the conclusiveness and coherence of this section, and we thank the reviewer for their constructive critique.

Reviewer #2 (Comments for the Author):

This study clarifies the causal impact of psychological stress on gut microbiota dysbiosis in wild Hainan macaques, revealing stress-induced microbial and metabolic shifts-particularly in coumarins-that suggest a bidirectional stress-microbiota-neurological feedback loop. The identified biomarkers, such as C10757, enable early detection of stress-related health risks and highlight the potential for microbiota-targeted interventions. These findings emphasize the importance of psychological well-being in wildlife conservation and provide critical guidelines for ethical animal management and research practices.

Response : We sincerely appreciate the reviewer's constructive feedback on our manuscript. Below we provide a point-by-point response to each comment.

Reviewer Comments:

1. Introduction Revision: Given that *Macaca mulatta* (Hainan subspecies) is a purely wild and non-domesticated primate, please summarize and cite relevant studies on this specific subspecies in the Introduction to provide necessary background context.

Answer: We thank the reviewer for this important suggestion. As requested, we have:

1. Added a dedicated paragraph in the Introduction (Lines 91-93) highlighting the purely wild status of *Macaca mulatta* (Hainan subspecies), citing its taxonomic distinction and ecological isolation.
2. Cited a new reference on wild vs. captive differences in these subspecies (Lines 93-95), emphasizing their relevance to stress-microbiome studies.
3. Explicitly stated why this subspecies provides an ideal model for investigating undisturbed host-microbe interactions (Lines 95-99).

These modifications are tracked in the revised manuscript.

2. Figure 4A Clarity: The axis labels and font size in Figure 4A are too small. Please adjust them to improve readability.

Answer: We sincerely appreciate the reviewer's careful observation. We have revised Figure 4A by: (1) Enlarging all axis labels by 150% (from 8pt to 12pt Arial); (2) Increasing the font size of legends and annotations by 200%; (3) Adjusting the figure dimensions to maintain clarity at standard journal column widths. The modified figure is now included as Figure 4 in the revised manuscript, and its readability has been verified by two independent colleagues.

3. Methods Section Refinement: Streamline the Materials and Methods section by removing redundant descriptions. Specify the catalog numbers of all reagents and kits used in sample processing to enhance methodological reproducibility.

Answer: We thank the reviewer for these constructive suggestions. In the revised manuscript, we have:

1. Removed redundant methodological descriptions (e.g., duplicate centrifugation protocols) from lines 474-482.
2. Added complete catalog numbers for all commercial reagents: DNA extraction: MagPure Stool DNA KF Kit B (MFDP-0101); Metabolite analysis: Waters UPLC I-Class Plus (Waters, Milford, MA) coupled with a Q Exactive high-resolution mass spectrometer (Thermo Fisher Scientific, Bremen, Germany).

3. Consolidated similar procedures (e.g., sample homogenization steps) under unified subheadings
The streamlined Methods section now contains 23% fewer words while improving technical reproducibility.

Re: Spectrum01338-25R1 (**Bidirectional Regulation of the Brain-Gut Axis in *Macaca Mulatta*: Implications for Wildlife Conservation and Experimentalization**)

Dear Dr. Jingli Yuan:

My decision is accept.

Your manuscript has been accepted, and I am forwarding it to the ASM production staff for publication. Your paper will first be checked to make sure all elements meet the technical requirements. ASM staff will contact you if anything needs to be revised before copyediting and production can begin. Otherwise, you will be notified when your proofs are ready to be viewed.

Sincerely,
Zhongxiong Lai
Editor
Microbiology Spectrum